# Influence of the Human Field of View on Visual and Non-Visual Quantities in Indoor Environments

**Johannes Zauner** [1,2,*] , **Kai Broszio** [3,4,*] **and Karin Bieske** [5]

1   3lpi Lichtplaner + Beratende Ingenieure mbB, Aidenbachstraße 52, 81379 Munich, Germany
2   Munich University of Applied Sciences, Lothstraße 34, 80335 Munich, Germany
3   Federal Institute for Occupational Safety and Health (BAuA), Friedrich-Henkel-Weg 1-25, 44149 Dortmund, Germany
4   Technische Universität Berlin, Einsteinufer 19, 10587 Berlin, Germany
5   Technische Universität Ilmenau, Ehrenbergstraße 29, 98693 Ilmenau, Germany; karin.bieske@tu-ilmenau.de
*   Correspondence: zauner@me.com (J.Z.); kai.broszio@alumni.tu-berlin.de (K.B.)

**Abstract:** The visual and non-visual effectiveness of light is often determined by measuring the spectrally weighed irradiance on the corneal plane. This is typically achieved using spectral irradiance or illuminance measurements, captured in a hemispheric (2π) geometry with a diffuser. However, the human binocular field of view (FOV) is not a perfect hemisphere, as it is occluded both upward and downward. Previous research on FOV-restricted measurements is limited, leaving the error from using hemispheric measurements for non-visual quantities undefined. In our study, we tackled this issue by designing and 3D printing FOV occlusions as attachments to spectral measurement devices. We took measurements with and without the occlusion in various laboratory (light from different directions) and real-world lighting situations (light typically from above). Our findings reveal a reduction of visual and melanopic values due to the FOV occlusion. These ranged from negligible to more than 60% in realistic scenarios. Interestingly, the reduction was consistent for both visual and melanopic parameters, as the distribution of light in the FOV was generally spectrally homogeneous. An exception occurred in a specific artificial laboratory situation, where the *melanopic daylight (D65) efficacy ratio* changed by more than a factor of 2 solely because of the FOV occlusion. Additionally, we observed that head orientation had a marked effect on all quantities measured. In conclusion, our results highlight the potential for substantial errors when solely relying on vertical, hemispheric measurements in experiments and non-visual lighting design projects. We encourage the (additional) use of FOV occlusion in eye-level measurements for typical viewing directions, and we are providing open-source 3D-print files to facilitate this practice.

**Keywords:** field of view; FOV; non-visual effect of light; non-image-forming effect of light; NIF; ipRGC; melanopsin; corneal illuminance; retinal illuminance; head orientation; radiance hood

## 1. Introduction

Light affects humans through both visual and non-visual pathways, stimulating photoreceptors in the retina [1]. While ideally one would measure the illuminance on the retina to predict these effects, practical constraints make this unfeasible [2]. Thus, corneal illuminance or spectral irradiance, typically measured at the position of the observer's eye and orthogonal to the gaze, serves as a proxy for non-visual effects. The standard way to measure irradiance involves a hemispheric approach, capturing the whole radiant flux from the front hemisphere. This method, known as 2π geometry, unfortunately includes light that would be blocked by human anatomy, such as the brow, eyelids, or cheekbone. A more accurate approach would only consider the light that can actually reach the eye's pupil through the binocular visual field of view (FOV). The CIE recommends this approach in Annex A.6 of the standard *CIE S 026:2018* [3]. Methods to achieve this include using

an occlusion or *radiance hood* to limit the detection areas of photometers and spectrora-diometers, or applying imaging measurements, e.g., from a luminance camera, followed by FOV-selective postprocessing [4–6]. However, these methods are not commonly used in current projects and studies, leading to potential inconsistencies and lack of standard-ization for non-image-forming effects of light (NIF). Our current understanding of the effect of FOV occlusion in real-world settings is limited. This study aims to bridge this gap by offering a method and data from a diverse set of scenarios to explore the differences between traditional $2\pi$ measurements and more precise FOV-focused approaches.

The human binocular visual field results from the combination of the left and right eye's monocular fields. Anatomical conditions, such as the eye's positioning in the orbit (eye socket) and eyelid placement, limit the possible visual field without head or eye movements [7]. Figure 1a illustrates the overlapping visual fields in a hemisphere for a simplified set of eyes (shown in red and blue), where the binocular visual field comprises all filled areas. This diagram is based on Guth's representation [8], with similar depictions found in Taylor [9]. It is crucial to note that individual visual fields can vary widely due to factors such as age, health, gender, and even stimulus attributes including size, luminance, and contrast. For example, the FOV in bright outdoor scenarios can be smaller than in typical indoor settings, e.g., due to eyelid placement [3]. The areas in Figure 1a thus provide general orientation rather than definitive measurements.

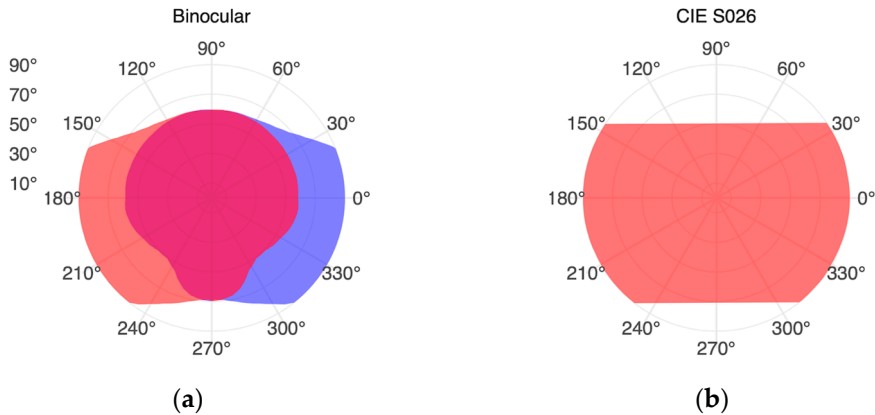

**Figure 1.** Representations of the human binocular field of view (FOV). (**a**) The overlapping visual fields of the left (red) and right (blue) eyes, forming the combined FOV. Data digitized based on Guth's representation [8]. (**b**) Recommended cutoff angles for the FOV in accordance with *CIE S 026:2018* [3], set at +50° and −70°, applicable for indoor environments with low light levels.

The standard *CIE S 026:2018* [3] provides recommendations concerning incidence angles grounded in the FOV. According to this standard, in indoor settings with low illuminance, the vertical field of view is considered to span up to +50° and down to −70° in relation to the line of sight. The monocular horizontal view is partially obstructed by the nose. However, for a binocular perspective, the horizontal field of view is considered to be about ±90° [3], as depicted in Figure 1b. It is crucial to highlight that the visual field exceeds beyond 180° in the horizontal direction. As Strasburger [10] discusses in *Seven Myths on Crowding and Peripheral Vision*, measurements by Rönne [11] indicate the horizontal FOV can extend up to ±107° or 214° in total. Yet, many publications cap it at ±90°, likely due to constraints related to measurement technology or stimulus presentation [10].

Regardless of how one defines the FOV, notable discrepancies surface between light incident on the cornea and measurements from the $2\pi$ geometry. Even with identical vertical illuminance, the spatial arrangement of light sources can cause significant variations in the portion of light falling within the FOV [4]. For instance, a high luminance light source might significantly impact vertical illuminance measurements but may be positioned outside the FOV. A practical example would be standard office lighting, where lights are placed high up at the ceiling. The given scenario would systematically overestimate the non-

visual stimulus when measured with a 2π geometry. This effect is further amplified when considering that in offices, people typically spend most of their time working with their heads slightly tilted downward, looking at the screen, and thus excluding even larger ceiling areas with lights from their FOV [12].

Such real-world scenarios starkly contrast with experimental setups found in key NIF research [1], summarized in Table 1. Many of these influential studies form the foundation of our current guidelines on light exposure during the day, evening, and nighttime [13]. In these studies, light stimuli are often presented as Ganzfeld illumination or centrally in the field of view, ensuring the light is primarily within the observer's FOV. Consequently, for these setups, the 2π-measurement of illuminance at the eye level probably provides an accurate representation of light intake.

**Table 1.** Summary of light stimulus characteristics in key NIF research studies. This table outlines the position (and size) of the light stimuli in some of the most significant studies in NIF research. These studies were examined by Brown (1) to demonstrate that "*melanopic illuminance defines the magnitude of human circadian light responses under a wide range of conditions*".

| Citation | Light Incidence [1] |
|---|---|
| Cajochen et al. 2000 [14] | *Not stated* |
| Zeitzer et al. 2000 [15] | Ceiling Mounted Lights [2] |
| Brainard et al. 2001 [16] | Ganzfeld Dome |
| Thapan et al. 2001 [17] | Ganzfeld Dome |
| Wright and Lack 2001 [18] | Low-Central, 20° visual angle |
| Revell and Skene 2007 [19] | Ganzfeld Dome |
| Brainard et al. 2008 [20] | Ganzfeld Dome |
| Gooley et al. 2010 [21] | Ganzfeld Dome |
| Revell et al. 2010 [22] | Ganzfeld Dome |
| Santhi et al. 2010 [23] | Central, Light Box |
| Papamichael et al. 2012 [24] | Ganzfeld Dome |
| Chellapa et al. 2014 [25] | Ganzfeld Room |
| Ho Mien et al. 2014 [26] | Ganzfeld Dome |
| Najjar et al. 2014 [27] | Ganzfeld Dome |
| Brainard et al. 2015 [28] | Central, 63° viewing angle |
| Rahman et al. 2017 [29] | Wall mounted Lights |
| Hanifin et al. 2019 [30] | Central, 63° visual angle |
| Nagare et al. 2019 [31] | Central, 40° viewing angle |
| Phillips et al. 2019 [32] | Ceiling mounted Lights |

[1] We use the term *Ganzfeld Dome* in all cases where it applies. Original descriptions vary, but fall in the same category [24]. [2] It is not clear from the publication whether the luminaires were part of the 2π-measurement geometry, but it is suggested that they are not [15].

As the field of NIF research advances, transitioning from controlled artificial setups (Table 1) to more realistic configurations [33,34] and field studies [35–39], we anticipate the impact of FOV occlusion to grow. Without proper consideration, FOV occlusion may introduce unexpected variance, complicating the replication of study setups, comparisons across publications, or the implementation of recommended stimulus intensities in lighting design. Despite its significance, current literature offers scant insight into the potential discrepancies between established measures of both visual and non-visual stimulus intensity and their FOV-occluded counterparts.

To address this gap, our study employed FOV occlusions designed in line with the standard *CIE S 026:2018* [3]. This design allowed us to more accurately measure the actual corneal illuminance or irradiance that reaches the eye. The occlusions were 3D printed from matte-black plastic, and special adapters were fabricated to facilitate compatibility with commercially available models of spectroradiometers. The FOV occlusions can also be used with photometer measurement heads.

In this exploratory, multicentric experiment, we assessed the influence of FOV occlusion on spectral irradiance and its derived quantities across diverse scenarios and

lighting conditions. Our goal was to gauge the relevance, magnitude, and range of the impact of FOV occlusion on these quantities. Consequently, we conducted measurements in 60 scenarios across 8 distinct projects, both with and without an FOV occlusion. The projects spanned artificial and realistic lighting situations in laboratory experiments and real-world workplaces.

As a commitment to further research, we have made the 3D print files for the FOV occlusion publicly accessible [40]. Our intention is to encourage other researchers to utilize them for non-visual light measurements and to report their findings along with established parameters. Over time, this collaborative approach will foster a more comprehensive understanding of the effects of FOV occlusion, well beyond the initial exploratory analysis of scenarios presented in the following section.

## 2. Results

The full set of results for all 60 scenarios is illustrated in Figure 2 and detailed in the Supplementary Information S1. The results presented in Figures 3 and 4 encompass 20 of the 60 scenarios measured across 8 distinct projects (A–H); these 20 scenarios were selected based on their relevance and are numbered (1–20). A summary of each project's result will be detailed in the following subsection. Additional insights on individual projects, including project descriptions, artificial lighting details, light settings, associated publication links, and extra images can be found in Appendix A. The projects are narratively organized, beginning with real or realistic workplaces (Project A–F) and followed by artificial laboratory settings (Projects G–H).

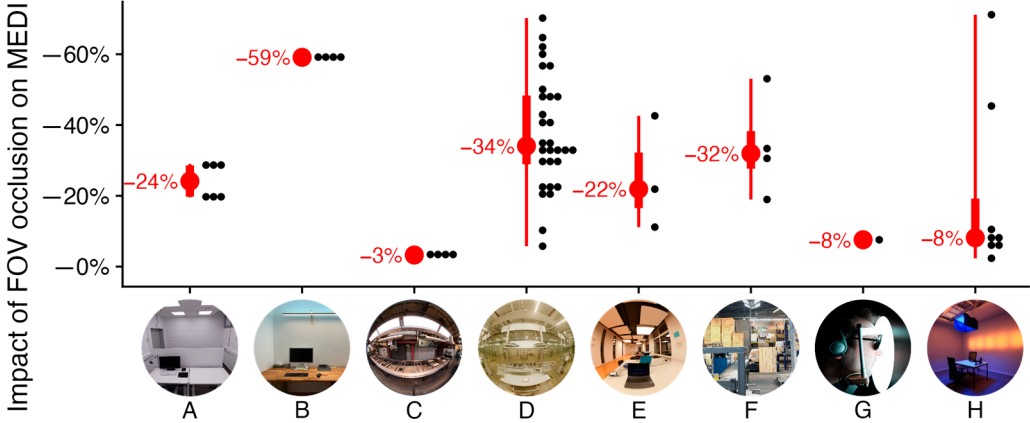

**Figure 2.** Summary across projects A–H displaying the relative reduction in non-visual stimulus intensity (MEDI) when applying the FOV occlusion versus the conventional method. Red dots represent the median values, with their numerical values to the left. The broader red bars signify the interquartile range (25–75%), while the narrower red lines indicate the range of recorded values. Black dots correspond to individual measurements and are aggregated in 2% bins for visualization.

All measurements were conducted at the observer's eye position and are noted as vertical unless specified otherwise. It is crucial to mention that all scenarios were restricted to artificial light, with no daylight involved, even though some scenarios may be named after different times of the day. The gathered spectral irradiance data were used to calculate key quantities and determine the relative differences attributable to the FOV occlusion (details can be found in Section 4, Materials and Methods). The analysis encompasses illuminance as a visual quantity and *Melanopic Equivalent Daylight (D65) Illuminance* (MEDI) [3] as a measure for the non-visual stimulus strength. Additionally, the *Melanopic Daylight (D65) Efficacy Ratio* (MDER) [3] represents the fraction of non-visual to visual quantity.

| | rSPD[1] | Name | Illuminance | MEDI[2] | MDER[3] |
|---|---|---|---|---|---|
| **A. Realistic Office Lab Study** | | | | | |
| 1 |  | Warm low | *2π:* 296 lx *FOV:* 212 lx **−28%** | *2π:* 141 lx *FOV:* 101 lx **−28%** | *2π:* 0.476 *FOV:* 0.476 **0%** |
| 2 |  | Cold low | *2π:* 298 lx *FOV:* 213 lx **−29%** | *2π:* 300 lx *FOV:* 214 lx **−29%** | *2π:* 1.007 *FOV:* 1.005 **0%** |
| 3 |  | Cold bright | *2π:* 830 lx *FOV:* 592 lx **−29%** | *2π:* 838 lx *FOV:* 595 lx **−29%** | *2π:* 1.010 *FOV:* 1.005 **0%** |
| **B. Home Office Workplace** | | | | | |
| 4 |  | Morning | *2π:* 450 lx *FOV:* 184 lx **−59%** | *2π:* 426 lx *FOV:* 171 lx **−60%** | *2π:* 0.947 *FOV:* 0.927 **−2%** |
| 5 |  | Daytime | *2π:* 390 lx *FOV:* 161 lx **−59%** | *2π:* 324 lx *FOV:* 131 lx **−60%** | *2π:* 0.831 *FOV:* 0.812 **−2%** |
| 6 |  | Night | *2π:* 139 lx *FOV:* 58 lx **−58%** | *2π:* 55 lx *FOV:* 23 lx **−59%** | *2π:* 0.397 *FOV:* 0.396 **0%** |
| **C. Industry Field Study (Machine Workplace)** | | | | | |
| 7 |  | Morning | *2π:* 449 lx *FOV:* 433 lx **−3%** | *2π:* 412 lx *FOV:* 399 lx **−3%** | *2π:* 0.918 *FOV:* 0.920 **0%** |
| 8 |  | Night | *2π:* 156 lx *FOV:* 150 lx **−4%** | *2π:* 72 lx *FOV:* 70 lx **−3%** | *2π:* 0.463 *FOV:* 0.466 **1%** |
| **D. Industry Workplace** | | | | | |
| 9 |  | Without Task Lighting | *2π:* 195 lx *FOV:* 79 lx **−59%** | *2π:* 137 lx *FOV:* 55 lx **−60%** | *2π:* 0.703 *FOV:* 0.690 **−2%** |
| 10 |  | With Task Lighting | *2π:* 270 lx *FOV:* 160 lx **−41%** | *2π:* 182 lx *FOV:* 104 lx **−43%** | *2π:* 0.673 *FOV:* 0.647 **−4%** |

2π = Hemispheric Measurements, FOV = Measurements with the Field of View occlusion, **Impact of occlusion in bold**
[1] Relative Spectral Power Distribution (rSPD), Hemispheric (2π) Measurement in stark color, FOV in faded color
[2] Melanopic Equivalent Daylight (D65) Illuminance (MEDI)
[3] Melanopic Daylight (D65) Efficacy Ratio (MDER)
[4] This viewing position equals the spectral measurement position

**Figure 3.** Results for part one of the selected projects (Project A–D, Scenarios 1–10).

| | rSPD[1] | Name | Illuminance | MEDI[2] | MDER[3] |
|---|---|---|---|---|---|
| **E. Learning Space** | | | | | |
| 11 [4] |  | Morning | *2π:* 261 lx<br>*FOV:* 232 lx<br>**−11%** | *2π:* 242 lx<br>*FOV:* 215 lx<br>**−11%** | *2π:* 0.927<br>*FOV:* 0.927<br>**0%** |
| 12 [4] |  | Daytime | *2π:* 209 lx<br>*FOV:* 164 lx<br>**−22%** | *2π:* 128 lx<br>*FOV:* 100 lx<br>**−22%** | *2π:* 0.612<br>*FOV:* 0.610<br>**0%** |
| 13 [4] |  | Evening | *2π:* 147 lx<br>*FOV:* 87 lx<br>**−41%** | *2π:* 54 lx<br>*FOV:* 31 lx<br>**−43%** | *2π:* 0.367<br>*FOV:* 0.356<br>**−3%** |
| **F. Industry Field Study (Packaging Workplace)** | | | | | |
| 14 |  | Morning | *2π:* 460 lx<br>*FOV:* 206 lx<br>**−55%** | *2π:* 322 lx<br>*FOV:* 151 lx<br>**−53%** | *2π:* 0.702<br>*FOV:* 0.736<br>**5%** |
| 15 |  | Daytime | *2π:* 225 lx<br>*FOV:* 154 lx<br>**−31%** | *2π:* 158 lx<br>*FOV:* 110 lx<br>**−31%** | *2π:* 0.704<br>*FOV:* 0.712<br>**1%** |
| 16 |  | Night | *2π:* 153 lx<br>*FOV:* 120 lx<br>**−22%** | *2π:* 97 lx<br>*FOV:* 79 lx<br>**−19%** | *2π:* 0.633<br>*FOV:* 0.654<br>**3%** |
| **G. Halfdome Ganzfeld Lab Setup** | | | | | |
| 17 |  | Halfdome | *2π:* 1,982 lx<br>*FOV:* 1,829 lx<br>**−8%** | *2π:* 2,323 lx<br>*FOV:* 2,147 lx<br>**−8%** | *2π:* 1.172<br>*FOV:* 1.174<br>**0%** |
| **H. Artificial Office Lab Study** | | | | | |
| 18 |  | LS1 | *2π:* 276 lx<br>*FOV:* 254 lx<br>**−8%** | *2π:* 164 lx<br>*FOV:* 90 lx<br>**−45%** | *2π:* 0.594<br>*FOV:* 0.353<br>**−41%** |
| 19 |  | LS2 | *2π:* 178 lx<br>*FOV:* 126 lx<br>**−29%** | *2π:* 256 lx<br>*FOV:* 238 lx<br>**−7%** | *2π:* 1.438<br>*FOV:* 1.889<br>**31%** |
| 20 |  | LS3 | *2π:* 181 lx<br>*FOV:* 117 lx<br>**−35%** | *2π:* 385 lx<br>*FOV:* 111 lx<br>**−71%** | *2π:* 2.127<br>*FOV:* 0.949<br>**−55%** |

2π = Hemispheric Measurements, FOV = Measurements with the Field of View occlusion, **Impact of occlusion in bold**
[1] Relative Spectral Power Distribution (rSPD), Hemispheric (2π) Measurement in stark color, FOV in faded color
[2] Melanopic Equivalent Daylight (D65) Illuminance (MEDI)
[3] Melanopic Daylight (D65) Efficacy Ratio (MDER)
[4] This viewing position equals the spectral measurement position

**Figure 4.** Results for part two of the selected projects (Project E–H, Scenarios 11–20).

*2.1. Project Results for Scenarios 1–20*

2.1.1. Project A: Realistic Office Lab Study, Scenarios 1–3

The FOV occlusion leads to a sizable reduction of the visual and non-visual quantities (−28% to −29% reduction), with little to no variation between scenarios. The occlusion is further nonspecific to spectral wavelength and thus MDER (0% change).

2.1.2. Project B: Home Office Workplace, Scenarios 4–6

The FOV occlusion reduces the visual and non-visual quantities by more than half (−58% to −60%). Further, even though the directionality of light (and spectrum) changes, we see the exact same behavior as in project A, i.e., very little variation of occlusion impact between scenarios. This can be attributed to the luminaire being mounted quite high above the observer, outside the FOV. Distributional changes between scenarios thus happen outside the FOV and have no impact on the proportion of light within to outside (changes in MDER 0 to −2%).

2.1.3. Project C: Industry Field Study (Machine Workplace), Scenarios 7–8

The FOV occlusion leads to a negligible reduction of the visual and non-visual quantities (−3% to −4%). Further, even though the directionality of light (spectrum) changes, we again see very little variation of occlusion impact between scenarios. Both aspects can be attributed to the workplace lighting, which is the dominant illuminant and within the FOV of the observer, while all surrounding areas are quite dark. Almost all light thus comes from within the FOV and changes of spectrum and light distribution do not matter for the FOV occlusion (changes in MDER 0 to 1%).

2.1.4. Project D: Industry Workplace, Scenarios 9–10

The FOV occlusion leads to a very high reduction of the visual and non-visual quantities (−41% to −60%). Further, this is the first project where we see variation in the impact of FOV occlusion between scenarios. This is because the proportion of light within the FOV increases by turning on the individual task lighting (from 40% to 57%; see the relative spectral power distribution (rSPD) between scenarios 9 and 10). The spectral impact of the FOV between scenarios is still negligible, however (changes in MDER −2% to −4%).

2.1.5. Project E: Learning Space, Scenarios 11–13

The FOV occlusion leads to a range of small to high reduction of the visual and non-visual quantities (−11% to −43%), depending on the scenario. This is quite a variation and can be attributed to the changing light settings between panel light and spotlight fixtures. The spotlight fixtures above the observers' table are outside the FOV, and their increasing contribution to quantities from unoccluded measurements also changes the impact of the FOV occlusion quite dramatically. Whereas the dominant panel lights illuminate the vertical surfaces of the walls within the field of view in scenario 11, their relative contribution compared to the spotlights is gradually reduced over scenarios 12 and 13. Despite the variation of the impact of FOV occlusion, spectral differences remain negligible (changes in MDER 0 to −3%).

2.1.6. Project F: Industry Field Study (Packaging Workplace), Scenarios 14–16

The FOV occlusion again leads to a small to high reduction of the visual and non-visual quantities (−19% to −55%), but even though the different light settings follow the same principles as described in project E, the order of impact levels from FOV occlusion is reversed from high to small. In this case, the dynamic lighting is mounted higher, and the observer's position is on the periphery of the workspace. The singular dynamic light above the observer thus has a high impact on visual and non-visual quantities without an FOV occlusion, but not with (scenario 14). As the luminous flux coming from the dynamic light is gradually reduced across the day (scenarios 15 and 16), the static stock lighting in the FOV becomes more dominant. The impact of FOV occlusion is thus also gradually

reduced. Despite this variation of the impact of FOV occlusion, spectral differences remain negligible to small (changes in MDER 1% to 5%).

### 2.1.7. Project G: Halfdome Ganzfeld Lab Setup, Scenario 17

The FOV occlusion leads to a very small reduction of the visual and non-visual quantities (−8%), with no spectral differences (changes in MDER 0%).

### 2.1.8. Project H: Artificial Office Lab Study, Scenarios 18–20

The FOV occlusion leads to the largest variation in reduction of the visual and non-visual quantities (−8% to −71%). This is because the artificial laboratory setup is designed to evoke stark spectral contrasts, and here it pays to look at the graphs of rSPD in Figure 4.

Scenario 18 shows almost no change on the visual side (−8%), because the dominant source of brightness comes from the wall straight ahead. It contributes little, however, to short wavelengths of high melanopic efficacy. Most of those spectral parts come from the panel straight above, outside the FOV. MDER and MEDI are thus highly affected by an FOV occlusion (−41% and −45%, respectively).

Scenario 19 flips the spectra for wall lights and ceiling panel, and the effects are flipped as well. Illuminance is highly reduced by the FOV occlusion (−29%), but MEDI are not (−7%). This leads to an increase in MDER of 31%.

Scenario 20 is a variant of scenario 18, but with flipped luminous intensities between wall and ceiling luminaires. This means that the general effects between visual and non-visual quantities and MDER are the same, but since most of the light now comes from the ceiling panel (instead of the wall), FOV occlusion restricts proportionally more light overall. Thus, illuminance is reduced by 35%, MEDI by 71%, and MDER by 55%.

### 2.2. Impact of Head Orientation (Project D)

In project D, 14 head orientations were measured in 15° increments besides the view straight ahead (used in scenarios 9 and 10). Figure 5 summarizes the results for MEDI and all 15 orientations, when individual task lighting is turned on.

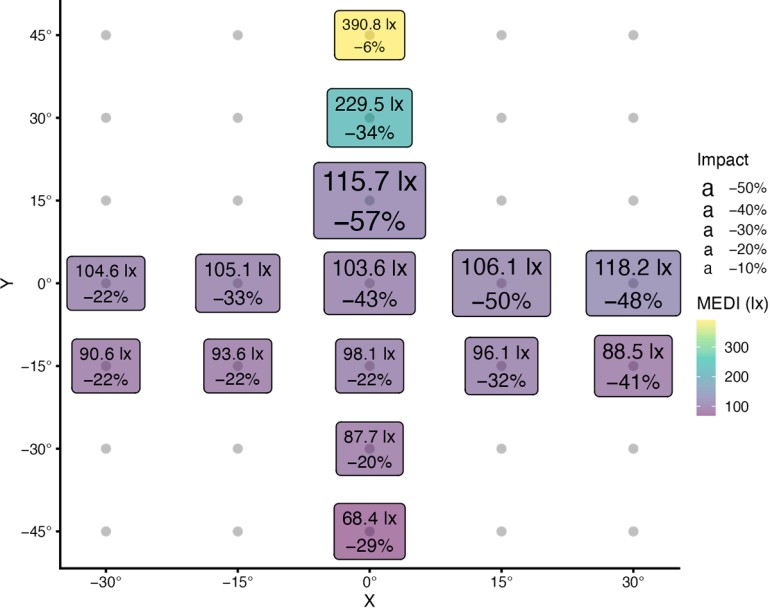

**Figure 5.** Graph of MEDI (lx) and the relative reduction of MEDI due to the FOV occlusion (impact, %), measured in 15° increments from a view straight ahead (0°/0°) in both horizontal (X) and vertical (Y) directions. The labels are scaled by MEDI (color) and impact (size). A small yellow label indicates a high MEDI value and a small impact, whereas a large blue label indicates a low MEDI value and a high impact. Measurements are from project D, with individual task lighting.

In the horizontal direction, there is considerable variation in the relative impact of FOV occlusion (−22% to −50%), but not MEDI (104 to 118 lx). This means that different proportions of light are occluded by the FOV (impact), but that this is offset by overall light levels. The same characteristic can be observed for horizontal steps with the head 15° lowered (Y:−15° in Figure 5).

In the vertical direction, MEDI values rise when looking up and fall when looking down (68 to 391 lx). Further, as the head is directed upwards by 15°, the impact of FOV occlusion becomes more prominent (−57% at Y:15°, compared to −43% at Y:0°), before it lessens with greater angles (−34% at Y:30°, and −6% at Y:45°). As the head is directed downward, the impact generally lessens, even if there is a small rise at the last step (−22% at Y:−15°, −20% at Y:−30°, and −29% at Y:−45°).

All of this may be explained by looking at the room layout. When looking straight ahead (or slightly upwards), workplace lighting straight above the observer will have a high impact on illuminance and MEDI, but its position is outside the FOV, thus the impact of FOV occlusion is high. As the observer looks further up, more and more ceiling lights move into the FOV, thereby decreasing the impact of FOV occlusion, but increasing light levels and MEDI at the eye. When looking downwards, ceiling lights will move outside not only the FOV but also the 2π geometry, which reduces the impact of occlusion, while reducing light levels and MEDI at the same time. As the observer looks further down, parts of the illuminated table surface move outside the FOV and the darker floor comes into the view. This leads to a slight increase in the impact of FOV occlusion and further decrease in MEDI.

## 3. Discussion

In the previous section, the effects of the human field of view on measurements regarding the corneal plane were demonstrated. Our goal was to determine whether and when FOV occlusion, according to the standard *CIE S 026:2018* [3], was relevant for indoor environments. To achieve this, we designed an FOV occlusion for spectral irradiance measurement devices and conducted measurements from eight distinct projects across laboratory and real-world workplace setups. Our findings provide a tentative answer to these questions.

To the best of our knowledge, there are no other publications that have tackled this topic with a focus on melanopic parameters. While our endeavor may seem trivial, as most conclusions can be deduced from the geometric principles of light measurement, FOV occlusion angles, and luminaires specifics, some of the results did surprise us. We now have a numeric basis for understanding the effect of FOV occlusion on the quantities we regularly work with for non-visual research.

Before diving into overarching project results, limitations of our study, and the further discussion of results in a wider context, we first want to provide a high-level summary:

- FOV occlusion is highly relevant for the visual and non-visual stimulus intensities in the context of realistic light-source positioning (mostly 20% to 60% reduction).
- Notable edge cases lead to a particularly high or low impact from occlusion (as low as −3% impact, to as high as −71%).
- FOV occlusion seems largely irrelevant for the spectral distribution (MDER). Only artificially constructed scenarios mattered in that regard, but it might also matter in spectrally diverse scenarios that were outside the scope of our projects.
- FOV occlusion is highly relevant in interaction with the head orientation (an impact as low as −6%, to as high as −57% for the same scenario, just by tilting the sensor).
- The significant variance of FOV impact between scenarios (Figures 3 and 4) and within scenarios (head orientation, Figure 5) prohibits use of a singular coefficient to correct for FOV based on standard illuminance/irradiance measurements alone.

## 3.1. Summarizing Project Results

Several more specific insights can be drawn. If not otherwise stated, impact scores for the FOV occlusion reference MEDI.

Dimming and tunable white:

- Changing the spectrum of light (most often through shifts in correlated color temperature (CCT)) across all lights in a scene the same way did not change the impact of FOV occlusion (scenarios 1–2, all about −29%). This does not mean, however, that changes in spectrum were not influential (see last point in Section 3.1).

    Changing the light output across all lights in a scene the same way did not change the impact of FOV occlusion (scenarios 2–3, all about −29%).

Direct/indirect lighting:

- Changing the direction of light and spectral distribution outside the FOV did not necessarily change the impact of FOV occlusion (scenarios 4–6, all about −60%), but see below in relation to spotlights and panels.
- Changing the direction of light and spectral distribution within the FOV did not necessarily change the impact of FOV occlusion (scenarios 7–8, all about −3%).

Installation height:

- Lights mounted at high vertical angles had a large impact on FOV occlusion (scenarios 4–6 and 9, up to −60%), but the impact can be reduced by wide beam lights in larger rooms, where more lights are visible at a lower viewing angle (scenarios 1–3, about −29%; scenario 12, about −22%).
- Low mounted or desk lighting reduced the impact on FOV occlusion (scenarios 7–8, about −3%; scenario 10: −43% compared to scenario 9: −60%)

Spotlight and panel-light fixtures:

- Mixing spot and panel-light fixtures in dynamic lighting produced beneficial effects through FOV occlusion. In scenario 11, the goal was to maximize MEDI: the dominant panel lights illuminated the room evenly and led to little FOV occlusion impact (−11%). In scenario 13, the goal was to minimize MEDI: the dominant spotlights illuminated mainly the desk surface, but the light source above the desk was also a large contributor to the standard $2\pi$ geometry. The FOV occluded this part, however, and thus reduced the MEDI further (−43%). The resulting dynamic MEDI range thus increased from the $2\pi$ geometry (242 lx/54 lx: factor 4.5) when adjusting for the FOV occlusion (215 lx/31 lx: factor 6.9).
- Mixing spot and panel-light fixtures in dynamic lighting produced detrimental effects through FOV occlusion. In scenario 14, the goal was to maximize MEDI: the dominant light panels were only above the observer, however, and did not illuminate the large hall (−53%). As MEDI values were supposed to become lower (scenarios 15–16), the panel lights were dimmed in favor of the spotlight fixtures; those did little, however, in lowering the light coming from the rest of the production hall (−19%). The resulting dynamic MEDI range was thus reduced from the $2\pi$ geometry (322 lx/97 lx: factor 3.3) when adjusting for the FOV occlusion (151 lx/79 lx: factor 1.9).

Head orientation (Section 2.2):

- (Near) vertical measurements led to the highest FOV occlusion impact (Y ± 0: −43; Y + 15: −57%), whereas large tilts up- and downwards reduced the impact considerably (Y + 45°: −6%; Y − 45°: −29%). We find it likely that this tendency is generalizable for typical workplace settings with overhead lighting (see 2.3 for more on this reasoning). While changes along the horizontal axis influenced FOV occlusion as well; we do not believe this can be generalized beyond our measurements.

Miscellaneous:

- Ganzfeld setups were small but not negligible in terms of the FOV occlusion impact (scenario 17, −8%), and real-world settings can have lower occlusion (scenarios 7–8, about −3%).
- The FOV occlusion reduced spectral irradiance by about the same amount regardless of wavelength (rSPD for scenarios 1–18). Thus, visual and non-visual quantities were also reduced in the same manner and changes in MDER are negligible.

This was not obvious to us as a common staple from the start. We believed that different spectral light or reflectance properties inside the FOV compared to outside might lead to significant deviations across the spectrum and thus MDER. While the surfaces in some projects were all neutral in their reflectance properties (scenarios 1–3, 7–10), in others they were not (e.g., wooden desk in scenarios 4–6; or wooden crates within the FOV in scenarios 14–16). However, only when artificially forcing strong differences in spectral distribution did we see an effect (scenarios 18–20). Similar effects might happen in very colorful settings.

### 3.2. Limitations

There are several limitations of note when interpreting and generalizing the results from this study.

### 3.2.1. Opportunity Sample Selection

The chosen projects and subsequent scenarios defined an opportunity sample, derived from existing projects where we had the chance to perform the necessary measurements. While we believe the projects present as valuable starting points with diverse lighting solutions, we must emphasize that they may not be representative for general workplace environments. Consequently, we have refrained from calculating universal statistics such as overall mean, variance, or effect size. Special consideration should also be applied in situations with colorful surface materials or lights, as seen in scenarios 18–20.

### 3.2.2. Incidence Angles of Light

There is evidence to suggest that differing angles of light incidence in the eye can alter the non-visual effect of an otherwise identical stimulus. Our study's results cannot contribute to this specific line of investigation. Previous research, such as that by Lasko et al. [41], indicated a significantly higher melatonin suppression when 500 lx of light came from above the gaze compared to below. However, both stimuli originated within the FOV (upper vs. lower FOV). Our measurements, integrating over either the full FOV or the $2\pi$ hemisphere, failed to capture the directional nuances within the FOV. For example, project C's scenarios 6 and 7, where light direction changed, showed no alteration in the FOV occlusion impact, remaining at 3%.

### 3.2.3. Gaze Direction

The human eye's ability to rotate in the orbit, determining the gaze direction in a stricter sense, further constrains the FOV. This aspect has been recently demonstrated in a mathematical model and accompanying measurements by He and colleagues [42,43]. Our FOV occlusion only considers a straight-ahead gaze and therefore does not take into account potential variations in gaze direction (Figure 1). We intentionally avoided the term *gaze direction* in the results, using the more precise descriptors of *head orientation* or *horizontal view*.

### 3.2.4. Definition of Illuminance and Cosine Correction

Beyond the mere FOV occlusion, illuminance's definition brings about additional complexity. Illuminance evaluates incident light from the hemisphere using a cosine function [44]. As the incidence angle increases, the relative contribution to illuminance and MEDI decreases. However, the human eye does not follow a cos-adapted lux metered pattern, even though the overall curve shape of the dependency may be similar [44,45]. In

comparison to a cosine correction, the eye's optical components diminish the incident light at a given angle less than what the cosine correction would suggest. Our FOV occlusion only cut off angles outside the FOV but did not correct for angles within (see Supplemental Figure S3 for details). In a situation with a homogeneous luminance distribution in the visual field, this can lead to a disparity of up to 6% compared to the cosine weighed integral measurement of the complete hemisphere [46]. In real situations, these differences are likely larger, and neither our study nor standard measurements for visual and non-visual quantities account for these. The angles with the largest differences between the two curves (cos-adapted and eye) are at least shaded vertically by the FOV [45]. Some related work was carried out by Van Derlofske et al. (46) in 2000, considering the angle-dependent characteristic of the eye and FOV limitations, even enabling scotopic measurements in a subsequent improvement [46,47].

### 3.3. Further Discussion and Outlook

Previously explored solutions for the FOV occlusion in measurements have included specialized apparatus by Van Derlofske and colleagues [46,47], using specially adapted optics in front of a photometer. Another solution that we only briefly touched upon in the introduction is based on luminance measurement cameras, as described, e.g., by Broszio et al. (4) and Babilon et al. (5). Both methods, however, carry disadvantages including high costs (Van Derlofske: approx. USD 1000 for photopic and scotopic measurements; luminance camera: >> USD 10,000), limited spectral resolution, and limited availability (Van Derlofske: complex replica). The two-camera setup by He et al. [42,43] allows for an extended FOV larger than 180° in the horizontal direction, but further inflates cost and complexity. Other methods, such as those described by Knoop et al. (6), have at least potentially capabilities similar to those of luminance cameras by simply choosing the measurement angles of relevance for the FOV. However, it is likely that they suffer the same downsides. The upside for spatial resolution methods includes the potential to add correction mappings that account for the non-cosine-adapted eye (see above). Another benefit is the flexibility regarding the FOV used in analysis. The CIE, for example, offers two FOV occlusion angles based on the overall luminance level that determines the state of the eyelids (indoor/outdoor scenarios) [3], and many more FOV models are described in the literature [10]. Fixed spatial approaches (such as ours) require an additional occlusion and measurement for each FOV, instead of applying (multiple) FOVs after the fact in computation.

The method presented in this publication offers the advantage of spectral measurement if used with a spectroradiometer, but it can also be adapted to illuminance measurement devices. It is further low-cost (material costs in 3D printing should not exceed a few euros), easy availability (public print files [40], short 3D printing duration), and adaptable to different geometries using CAD software to construct other mounting mechanisms. This grants researchers in the field of non-visual effects of light the ability to describe lighting situations more precisely with minimal effort [6,48]. Even for those lacking access to a spectroradiometer, the FOV occlusion method can be applied to simple devices like photometers, offering valuable insights into corneal illuminance.

We initiated this publication by referencing key relevant studies on non-visual light effects of light that help us to define dose–response relationships and recommendations for stimulus levels across the day [1,13]. As Table 1 shows, these studies were conducted using predominantly Ganzfeld geometries or lights in the central FOV. Our results indicate minor (scenario 17: −8%) or negligible (scenarios 7–8) impacts on FOV occlusion in those cases, respectively (see Figures 3 and 4). Even this assumption might not hold, as Zauner et al. [49] report a 24% reduction of stimulus size when considering the FOV in the Ganzfeld dome of their study. This means there is at least some variation in the light distribution of Ganzfeld conditions. In practical situations with ceiling mounted lights, however, the deviation to the dominant conditions detailed in Table 1 is much higher. Our measurements show a significant reduction of up to 60% for illuminance and MEDI when considering the FOV

(at horizontal view and realistic settings, see Figures 3 and 4). These results are consistent with the still sparse literature on this topic [5,46] and have a high relevance towards study design and replicability.

Consider an example, where a lighting situation in an experiment was intended to achieve 250 lx MEDI, but only 150 lx MEDI can reach the eye (this would equal an FOV occlusion impact of −40%). Without appropriate corrections and only the 2π measurement reported, the significant variance shown in Section 2 remains hidden, e.g., in a deduced dose–response relationship. If, on the other hand, corrections are made purely through luminous flux to achieve the target value, approximately 1.7 times the luminous flux would have to be applied. This raises the (unrestricted) vertical MEDI value to around 420 lx, which might be an acceptable solution for an experimental setting. However, it is hardly practical as a general recommendation, as it is energy consuming and might lead to other undesirable side effects. Rather, it should be solved through the types and arrangement of light sources, at least in real-world settings.

Another relevant aspect is the importance of head orientations. It seems that the standard vertical measurement direction (horizontal view) is particularly undesirable with realistic light settings. This is because it leads to a far greater error than other viewing directions, which are typically downwards at a monitor (about −15° tilt) or a workpiece on the desk (about −45° tilt), at least in our projects. Because the impact of FOV occlusion and MEDI change in a major way with the head orientation, we strongly recommend including typical viewing directions in addition to vertical measurements.

Finally, our results have other implications for (non-visual) lighting design, besides energy efficiency. Existing recommendations do mention the importance of light-source placement within the FOV (e.g., [50]), but as this aspect is not integrated in the standard MEDI measurements, it can be easily forgotten along the design process. The results of this study clearly show that even very bright light sources directly above the observers' eye point do not or only slightly contribute to corneal illuminance (see, e.g., scenarios 4, 9, 13, or 14). This can be advantageous for an evening lighting situation when a work surface needs to be illuminated, but the non-visual stimulus intensity at the eye should be limited (scenario 13). For activating situations such as morning hours, it is much more beneficial to bring light from the front or diagonally above the eye (scenario 11). In these cases, the FOV occlusion has only minimal influence. In general, this can be achieved through low-mounted luminaires and indirect lighting on walls or more distant ceiling surfaces. Considering the FOV occlusion in this way thus supports a targeted and energy-efficient non-visual stimulus for experimental and practical designs.

In conclusion, this study underscores the importance of FOV occlusion in determining the visual and non-visual stimulus intensity. Though our investigation can only be considered tentative in terms of how representative our settings are for real-world scenarios, the results strongly hint at a high relevance for the topic. We have also provided the means for others to extend this research with their own measurements [40]. Lastly, we believe that FOV occlusion should be considered as a mainline entry for future iterations of the standard *CIE S026* instead of the informative part, alongside such relevant factors as age [3]. This would encourage further adoption among researchers and practitioners.

## 4. Materials and Methods

### 4.1. FOV Occlusion

The FOV occlusion was constructed according to the standard *CIE S 026:2018* [3]. The aperture was designed so that the entire diffuser surface was shaded at the given angle. The aperture shape was then obtained by a line with the angle from a point on the edge of the diffuser to the corresponding point on the opposite side of the aperture's diameter. Figure 6a,b show some of these construction points. In Figure 6c,d, the resulting aperture is shown. Three-dimensional print files for the FOV occlusions and mounting clamps for Jeti spectrometers can be downloaded freely from an online repository [40]. The FOV occlusions were 3D-printed from matte-black plastic. Finally, we verified the correct

behavior of the FOV occlusion towards the cutoff angles through measurements of the relative illuminance from a narrow beam light from different directions (Supplementary Figure S3).

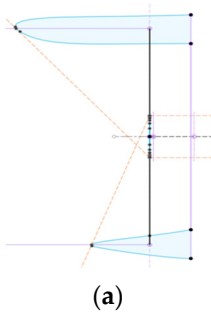 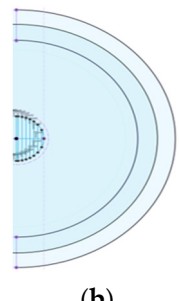 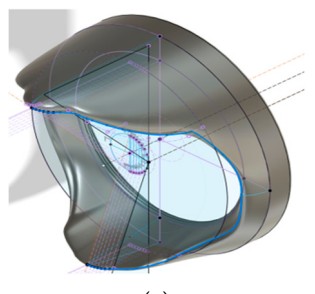 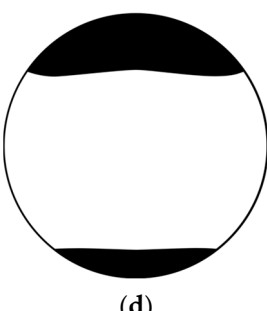

(**a**) (**b**) (**c**) (**d**)

**Figure 6.** Design schematics of the FOV occlusion. (**a**) Vertical section through the aperture construction with dashed orange +50° line and −70° line for limiting the upper and lower visual field. (**b**) Partial frontal view with selected construction points on the edge of the central diffuser. (**c**) Complete aperture in 3D view with selected construction lines. (**d**) A 180° fisheye-view schematic through the aperture of the FOV occlusion. White areas represent the FOV, black the occlusion from the 2π geometry. The reason why the upper and lower occlusion areas do not have straight cutoffs horizontally (compare Figure 6d to Figure 1b) is due to the aperture construction that shades the whole diffuser instead of just the central point.

For the projects B–F, a simpler preliminary version of the FOV occlusion was used (see Figure 7c). In practice, this is of little relevance, as both occlusions show about the same angle-dependent characteristics (see Supplementary Figure S3 for the comparison). Further, projects A, G, and H were measured with both variants, and the median absolute deviation for quantities between the two occlusions was small at 2% FOV occlusion impact. Due to the small differences and some projects being available exclusively with measurements from the preliminary aperture, we decided to include measurements from the prototype in the analysis. For reasons of reproducibility and transparency, this prototype is included in the online repository [40].

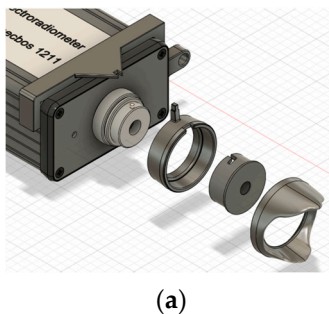 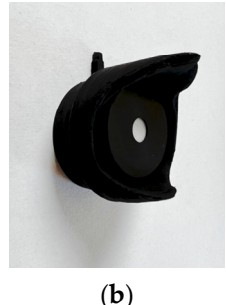 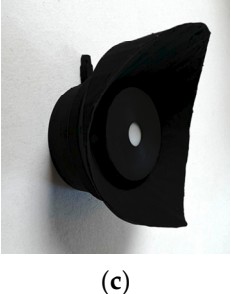

(**a**) (**b**) (**c**)

**Figure 7.** FOV occlusions: mounting. (**a**) Mounting principle of the occlusion on a *JETI specbos 1211*. A mounting clamp on the device attaches to a distance ring before the diffuser is attached. The FOV occlusion then attaches to the distance ring with a reproducible positioning. (**b**) Picture of the FOV occlusion with the distance ring and diffuser cap. (**c**) Same as (**b**), with the prototype FOV occlusion.

### 4.2. Measurement Apparatus

Spectral irradiance measurements with a 1 nm resolution were performed using a *JETI Specbos 1201* or *1211* spectroradiometer (*JETI Technische Instrumente GmbH*, Jena), with the *JETI LiVal V6.14.2* software running on a connected personal computer. The software also allowed for the calculation of illuminance and MEDI based on the measurement. MEDI were calculated according to the standard *CIE S 026:2018* [3]. The spectroradiometer's

relative measurement accuracy was 2%. Raw data and derived quantities were saved in comma-separated value (CSV) files and can be found in the Supplementary Information S2.

### 4.3. Projects and Measurements

All measurements in this study were collected from eight projects available to the authors. Each project consisted of a workplace situation that came from a range of laboratory or field studies. Measurements were taken with a spectroradiometer at the observers' eye level at a height of 1.2 m for seated workplaces and at a height of 1.6 m or 1.5 m for standing individuals. All measurements were taken twice, once without the FOV occlusion, and a second time with the occlusion. Appendix A gives a brief description of each project, the light and light settings, and the relevant observer. Only artificial light was used; no daylight is present in the measurements.

### 4.4. Data Analysis

We used the *R* software (Version 4.2.3) [51] for data analysis. Besides image and table generation, only a few analysis steps were performed. First, the *Melanopic Daylight (D65) Efficacy Ratio* (MDER) was calculated from photometric illuminance and MEDI, according to the standard *CIE S 026:2018* [3]. Further, each quantity (illuminance, MEDI, and MDER) consisted of a pair of measurements, one with the $2\pi$ geometry (unoccluded) and one with the FOV occlusion. An impact score of FOV occlusion was calculated for each pair with:

$$FOV\ Occlusion\ Impact_{i,\ Quantity} = Quantity_{i,rel_{FOV}} - 100\%, \tag{1}$$

$$Quantity_{i,rel_{FOV}} = \frac{Quantity_{iFOV}}{Quantity_{i2\pi}} \times 100\%. \tag{2}$$

The impact score can be read as an effect size and specifies in a percentage how much the FOV occlusion reduced the value of the given quantity in scenario *i*, compared to the respective $2\pi$ geometry. That is, an impact score of $-28\%$ for MEDI ($i:1$) states that MEDI values were 28% lower when using the FOV occlusion in scenario 1, compared to a standard measurement. Only in case of MDER were positive impact scores possible, as MDER is a relative quantity. The number given in each relative spectral power distribution plot (rSPD) of Figures 3 and 4 is the median *Spectral Power* $_{i,\lambda,rel_{FOV}}$ according to Equation (2), for wavelengths $\lambda$ between 380 nm and 780 nm. All scripts for data analysis and image generation are in the Supplementary Information S4.

**Supplementary Materials:** The following supporting information can be downloaded at: https://www.mdpi.com/article/10.3390/clockssleep5030032/s1, ZIP-Folder S1: All measurement results in Figures and as CSV data. ZIP folder S2: All raw measurement data and derived parameters. Figure S3: Verification results for cutoff angles of FOV occlusion. HTML document S4: Quarto file containing all scripts used for data analysis and image generation, alongside the respective results.

**Author Contributions:** Conceptualization, J.Z., K.B. (Kai Broszio) and K.B. (Karin Bieske); methodology, J.Z., K.B. (Kai Broszio) and K.B. (Karin Bieske); software, J.Z. and K.B. (Kai Broszio); validation, K.B. (Kai Broszio) and K.B. (Karin Bieske); formal analysis, J.Z.; investigation, J.Z., K.B. (Kai Broszio) and K.B. (Karin Bieske); resources, J.Z., K.B. (Kai Broszio) and K.B. (Karin Bieske); data curation, J.Z., K.B. (Kai Broszio) and K.B. (Karin Bieske); writing—original draft preparation, J.Z. and K.B. (Kai Broszio); writing—review and editing, J.Z., K.B. (Kai Broszio) and K.B. (Karin Bieske); visualization, J.Z.; supervision, J.Z., K.B. (Kai Broszio) and K.B. (Karin Bieske); project administration, J.Z., K.B. (Kai Broszio) and K.B. (Karin Bieske); All authors have read and agreed to the published version of the manuscript.

**Funding:** This research received no external funding.

**Institutional Review Board Statement:** Not applicable.

**Informed Consent Statement:** Not applicable.

**Data Availability Statement:** The data presented in this study are available in Supplementary Information S2. The 3D print files for the FOV occlusions are openly available from: https://doi.org/10.14279/depositonce-17076 (accessed on 01 August 2023) [40].

**Acknowledgments:** While no funding was given for the measurements (or any other part) of this study, the measurements would not have been possible without the projects themselves. All authors would thus like to thank the funders of and partners in those projects (see Appendix A for more information on the projects). KBr would like to thank Hannah Rolf for providing access to the lab and the possibility to conduct measurements of her study lighting settings. KBr would like to acknowledge the work by bachelor thesis student Maximilian Lutz (implementation of the 3D design; conduction of evaluation measurements of the FOV occlusions) and by student worker Noah Pütz (conduction of evaluation measurements of the FOV occlusions). Finally, we would like to thank two anonymous reviewers for their constructive and helpful comments.

**Conflicts of Interest:** The authors declare no conflict of interest.

## Appendix A. Project Descriptions

This section contains more information on the projects and their lighting design. The pictures can also be found at the Open Science Framework (OSF) project [52].

*Appendix A.1. Project A: Realistic Office Lab Study, Scenarios 1–3*

The measurements in the first project came from a laboratory study on non-visual effects over the course of a workday in an office-like environment [53,54]. Luminaires integrated into the ceiling allowed for control of the spectral composition of light. The three investigated light situations differed in vertical illuminance and MEDI value. The light situations "Warm low" and "Cold low" had the same vertical illuminance of 300 lx at the eye. However, they differed by approximately a factor of 2 in the MEDI value. The "Cold bright" light situation increased the vertical illuminance and the MEDI value compared to the "cold low" situation by 2.8 times. Measurements were taken from the observers' perspective seated at the desk, 1.2 m above floor level. For every scenario, measurements were taken with a horizontal view (Figure 3) and with an 20° angle downwards from the horizontal plane (not shown in Figure 3).

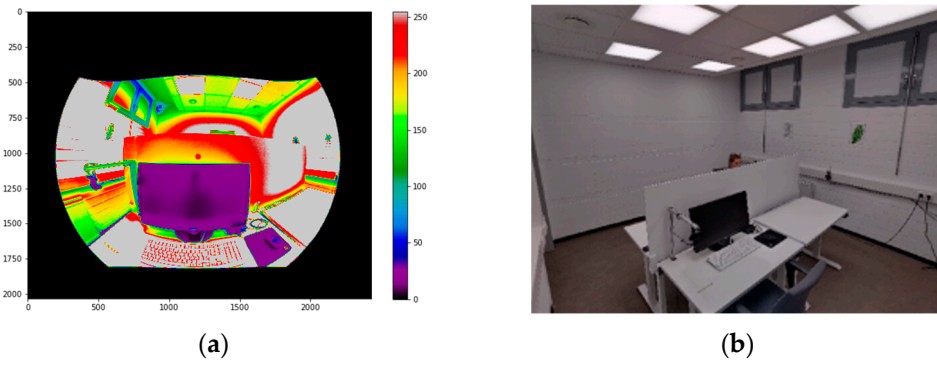

(**a**)  (**b**)

**Figure A1.** (**a**) Fisheye-view (false color luminance in cd/m$^2$) for the "Cold bright" situation, including the FOV occlusion. (**b**) Additional perspective.

*Appendix A.2. Project B: Home Office Workplace, Scenarios 4–6*

A self-constructed luminaire from a teaching project used in a real-world home office allowed settings for different lighting scenes depending on the time of day. The lighting scenes differed in light output, light color, and light direction from direct and indirect light components in the linear luminaire. Morning (direct: 7000 K, high luminous flux (LF); indirect: 5000 K, high LF), daytime (direct: 4000 K, normal LF; indirect: 7000 K, normal LF), evening (not shown in Figure 4, direct: 4000 K normal LF; indirect: 2700 K low LF), and night (direct: 2700 K, normal LF; indirect: 2700 K, low LF). The directionality of light thus

changed between scenarios. For example, in scenario 4, high LF came from the ceiling and with a cooler light spectrum than from the direct component, whereas in scenario 6, the LF of the indirect component was minimal and had the same warm white light spectrum as the direct component. Measurements were taken with a horizontal view from the observers' perspective sitting at the desk, 1.2 m above the floor.

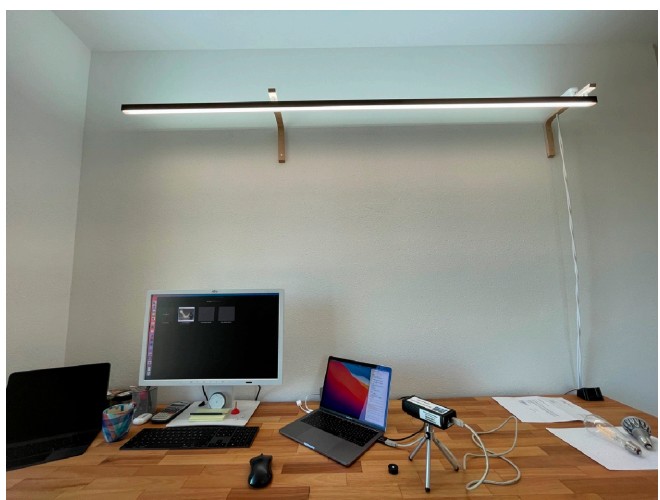

**Figure A2.** Full view of the home office workplace.

*Appendix A.3. Project C: Industry Field Study (Machine Workplace), Scenarios 7–8*

In this project, carried out jointly with the German Social Accident Insurance Institution for the Administrative Sector (vbg), an industrial workplace for quality control in a continuous three-shift operation was consistently designed and executed with a focus on the non-visual effects of light. A detailed overview of the project and the lighting is given in Zauner and Plischke (37). In brief, the specially developed *Drosa* luminaire allowed the adjustment of wing-panel luminaires in spectrum and brightness, as well as the brightness of 4000 K spotlight fixtures that shone directly on the workplace. Depending on the time of day, transitions were made between different lighting scenes. In all cases, the direct illumination of the spots provided horizontal illuminance on the visual task of at least 900 lx. In the morning scenario, mainly the wing-panel luminaires were active with a daylight white light spectrum (6500 K). In the daytime scenario (not shown in Figure 3), the wing-panel luminaires were slightly dimmed, and the CCT was adjusted to 4000 K. During the night scenario, the wing-panel luminaires were highly dimmed and mainly provided a general illumination level with a warm white light color (3000 K). In the second half of the night, the illuminance was gradually increased until the end of the night (not shown in Figure 3). The measurements were carried out from the observers' position at a typical workstation in quality control. This was a machine workplace in a standing position, with the eye level 1.5 m above the floor and a horizontal view.

*Appendix A.4. Project D: Industry Workplace, Scenarios 9–10*

In this study, a laboratory room with a workplace was set up and designed to resemble real-world conditions, based on measurement results at assembly workplaces in an industrial hall. The real-world mockup further extended to the LED light for general and task specific lighting (individual task lighting). Mirrored walls on three sides of the laboratory room created the impression of a larger room geometry. The settings for the lights were chosen so that comparable values to those of the assembly workplaces were measured in the working plane and vertically in the eye position. The dimmable ceiling light had a variable color temperature between 2700 K and 6500 K, and was set to 4500 K. The luminous flux of the luminaires was adjusted so that the horizontal illuminance was at 500 lx. The non-dimmable task specific light (termed *APL* in the data) had a static color

temperature of 4000 K and increased illuminance on the horizontal work plane by 450 lx. Even though the accompanying investigation of this workplace involving humans was a laboratory study, for this publication we categorized it as a field study. This is because the lighting in the laboratory room can be considered a 1:1 mockup of an existing real-world workplace, not an abstraction or generalization of industry workplaces. Measurements were taken at the observers' point of view, with a height of 1.6 m above the floor. A total of 30 measurements were taken in this project. Half of those measurements were with APL turned on, and half turned off. The measurements further varied the head orientation vertically and horizontally in 15° increments. An overview of all head orientations is found in Figure 5. Figure 3 only shows results for a 0°/0° view.

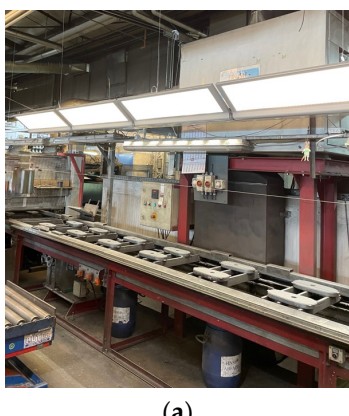

(**a**)

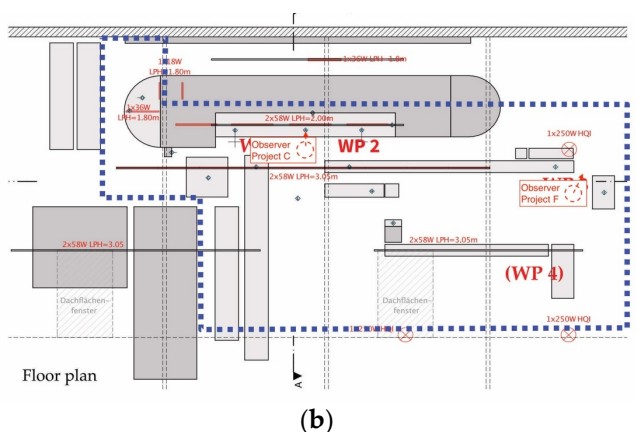

(**b**)

**Figure A3.** (**a**) Sideview of the workplace lighting. (**b**) Floor plan with the relevant observer position C in the middle.

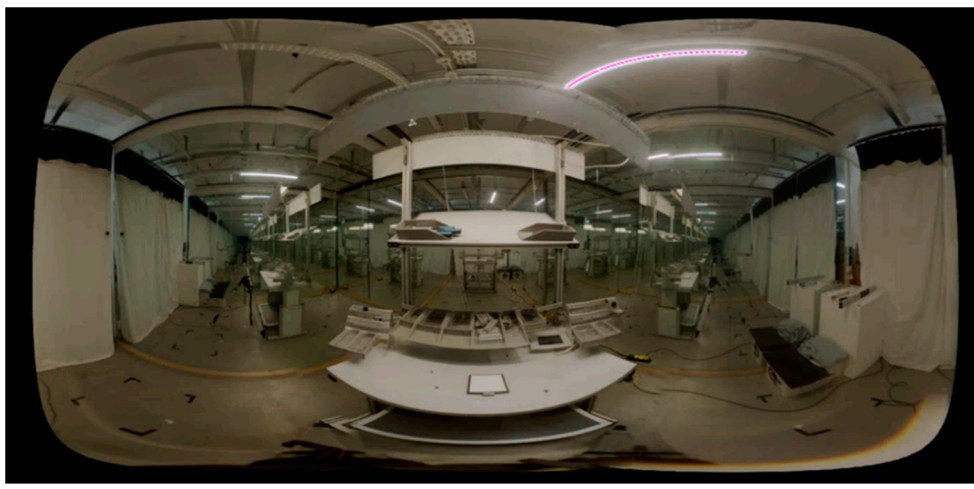

**Figure A4.** Panorama view of the laboratory recreation of the real workplace.

*Appendix A.5. Project E: Learning Space, Scenarios 11–13*

As part of the "Learning Space of the Future" project, an existing learning space at Munich University of Applied Sciences was equipped with a dynamic lighting solution. A detailed overview of the project and the lighting is given by Zauner [55]. In brief, in addition to dimmable and spectrally variable panel luminaires, spotlight fixtures on the ceiling above the table could also be controlled. Depending on the time of day, transitions between various lighting scenarios were made, all of which led to a horizontal illuminance of 500 lx on the work surface. In the morning scenario, predominantly panel luminaires were active, and all luminaires had a CCT of 7000 K. In the daytime scenario, the panel

luminaires were slightly dimmed, and the CCT was adjusted to 4000 K. The evening scenario was dominated by the spotlights, while the panel luminaires provided little light output and mainly created a general illuminance level. All luminaires in this scenario had a warm white light color and were set to 2700 K. The measurement was carried out for a typical user workstation in the room, with a sitting observer's eye position 1.2 m above floor level and a horizontal view.

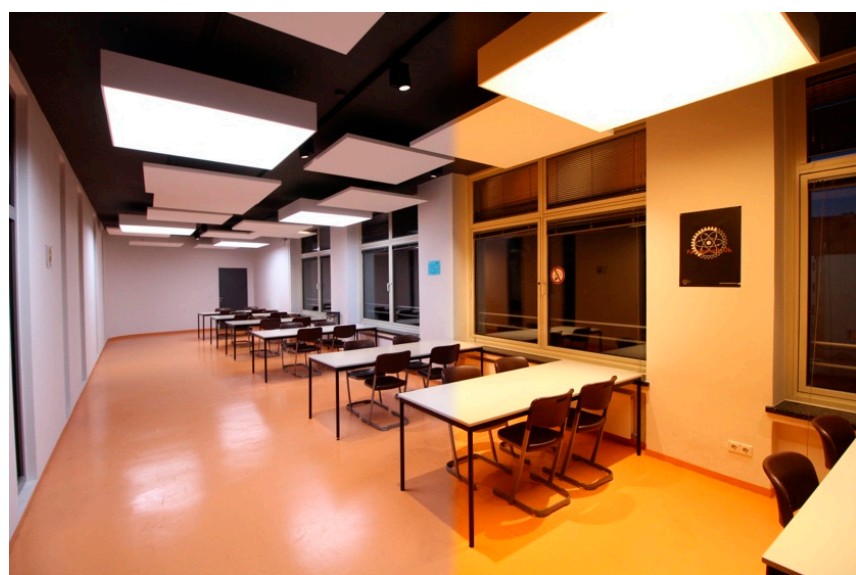

**Figure A5.** Morning/evening side view of the room.

*Appendix A.6. Project F: Industry Field Study (Packaging Workplace), Scenarios 14–16*

This project was in the same industrial workspace as project C [15], but located in the periphery. The general conditions for lighting and measurement were thus almost identical. One difference was the target illuminance level for the visual task at 300 lx. Another difference was that the lighting was mounted higher by about 1 m. The final difference was in the view at the worker's position (1.5 m above floor level, measured horizontally), which was open and overlooking the other areas in the industrial hall. This included the stock illumination in all other areas, which consisted of fluorescent tube lights.

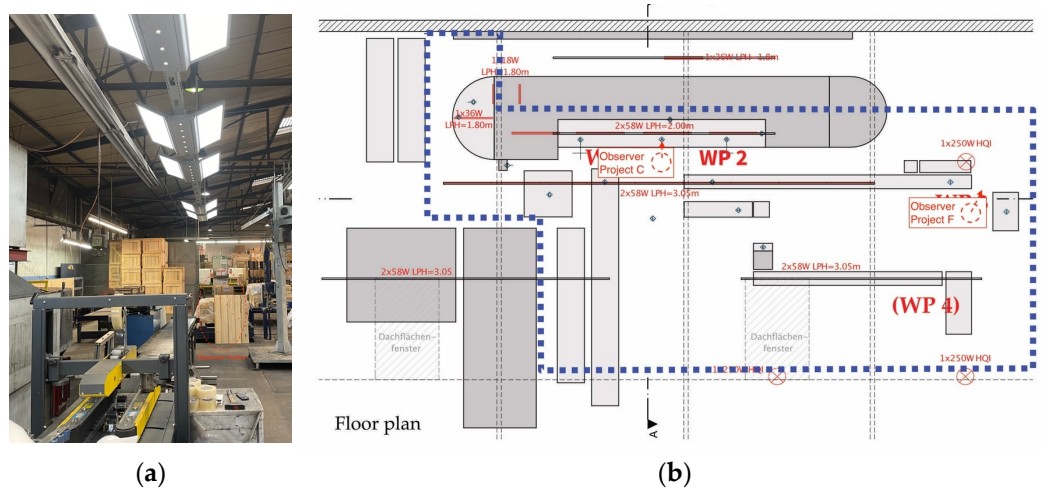

(**a**)                                 (**b**)

**Figure A6.** (**a**) Sideview of the workplace lighting. The red ellipse marks the observer position. (**b**) Floor plan with the relevant observer position F on the right.

As part of a laboratory study on non-visual effects of light, test stations with Ulbricht hemispheres were examined [56]. The hemispheres allowed for a 2π geometry of illuminance in the subject's field of view, i.e., the Ulbricht hemispheres were illuminated in such a way that a constant luminance prevailed inside. The test subjects looked into the opening of the hemispheres using a chinrest. Measurements were taken vertically at the eye level of the test subjects, which was centered on the hemisphere.

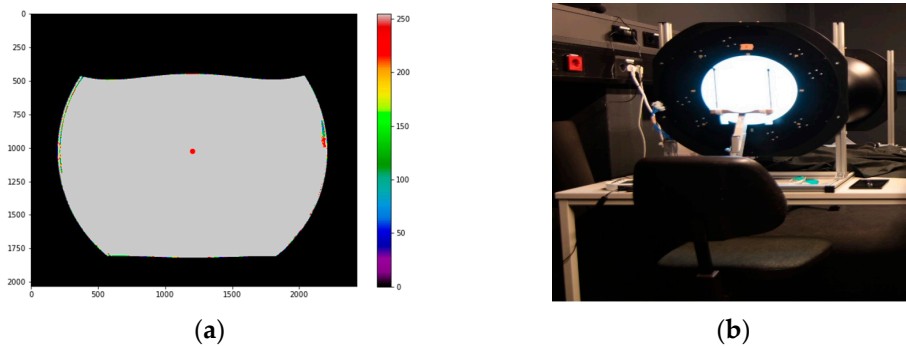

(**a**) (**b**)

**Figure A7.** (**a**) Fisheye-view (false color luminance in cd/m$^2$). (**b**) Outside view.

As part of a laboratory study on nocturnal non-visual effects of light, a test station was set up in a room that represented an office-like environment. Different lighting scenarios could be realized using ceiling and wall-integrated luminaires. The project investigated whether a lighting solution using the spectral properties and spatial arrangement of the luminaires could be designed to maintain melatonin secretion at night while simultaneously supporting the acute attention of night-shift workers [57]. Three lighting conditions, which differed in direction and spectral composition of the light, as well as a dim light condition (not shown in Figure 4), were examined. Light scene LS1 had warm white illuminated walls and dim blue light from a central ceiling panel above the observer. The ceiling panel was occluded sideways, so that it only shone light straight down at the observers' desk. LS2 had blue light on the wall and dim warm light from the ceiling panel. LS3 had dim warm white light at the wall and bright blue light from the ceiling panel. Measurements were taken at sitting height 1.2 m above floor level, with a horizontal view (Figure 4) and with an 20° angle downwards from the horizontal plane (not shown in Figure 4).

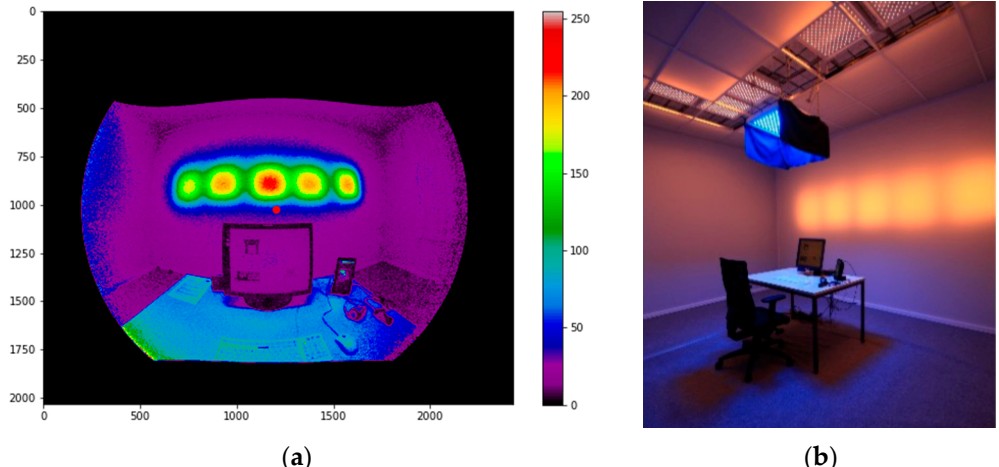

(**a**) (**b**)

**Figure A8.** (**a**) Fisheye-view (false color luminance in cd/m$^2$) for the "LS3" situation, including the FOV occlusion. (**b**) Additional perspective for LS3.

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
