# Peer review of "Influence of the Human Field of View on Visual and Non-Visual Quantities in Indoor Environments"

_2624-5175, doi:10.3390/clockssleep5030032_

Round 1

Reviewer 1 Report

The application of a standard Field of View (FOV) in the determination of light entering the eye for the purpose of analyzing circadian or other similar impacts of light on an occupant makes sense, although it still generally requires the selection of a fixed observer position and view direction that doesn’t fully represent the real-world exposure conditions of a person in that space.  While this procedure is quite logical, the value in this study is that it illustrates the impact of applying a FOV solution versus a more simplified hemispherical vertical illuminance at an observer position.  Other differences besides the FOV impact are also discussed.

The consideration of the range of different space and lighting conditions across the selected projects is quite helpful.  One area where this paper could be improved is in the documentation/specification of the lighting situation in each of the case studies.  For some of these cases, the view image is a hemispherical fish-eye view, which provides most of the required details, but for others, a standard camera photograph is provided that does not cover the entire hemisphere and important details are missing.  One example of this is the Home Office Workspace, where the FOV illuminance is less than half of that of the unobstructed vertical illuminance.  Clearly, as explained in Section 2.2.2, this is due to a ceiling-mounted luminaire that is outside the provided camera image.   It would be extremely helpful if the authors could revisit the spaces in A, B, F and H to re-photograph these spaces with a fisheye lens to provide consistent documentation across all of the test spaces. 

Line 137 – I don’t understand why the projects are only “(A-D)” in this sentence.

Line 139 -  I suggest that you list what important details can be found in the Appendix for these projects.

Line 308 – Change “High mounted lights” to something like “Lights mounted at high vertical angles”

Overall, I feel this paper documents the differences between the full hemispherical vertical illuminance and the application of a more limited FOV approach to quantifying the light that is incident at the eye, in terms of both magnitude and spectrum.  Future research, measurements, and simulations in this area are likely to require the FOV approach, so this paper is certainly timely and useful.

Reviewer 2 Report

The paper presents valuable research on the effects of field of view (FOV) occlusion in light measurements and its relevance for indoor environments. I personally think that the topic is very interesting, under-studied, and fits the scope of the journal. In lighting and chronobiology research, the impact of peripheral light is seldom investigated. The fundamental knowledge of peripheral vision goes back to older research studies (i.e., Poppel et al., Stiles-Crawford etc). So, it is good to see the topic picking up interest again.

About the manuscript: the introduction provides a good background, but it could benefit from a clearer statement of the specific research gap and contributions of this study. The abstract gives a good overview, but more specific details about the methods, results and key findings would be helpful. The discussion section provides informative insights. The paper emphasizes reproducibility, providing 3D print files and measurement apparatus details, which is commendable.

Overall, the paper makes a valuable contribution to the field, and with some refinements, it will be even more significant. The research is well-executed, and the authors' efforts to provide open-access resources for reproducibility are appreciated.

Some minor comments:

It might be a language issue, but the statement in lines 62-63 “[visual field] influenced by many factors, such as the observer (age, health, gender, experience, etc.) and the stimulus (size, luminance, contrast, etc.).” doesn’t make sense. Field of view does not change with stimulus or experience. It can change depending on the orientation and movements of the observer's head and eyes. I wonder if authors mean something else here.

The statement in lines 295-298 about the spectrum effects should be written with caution. Spectrum might be influential depending on the angle light rays entering the eye (yet again another understudied area). Also, CCT is not a precise measure/proxy for spectra, let alone circadian entrainment. Authors can look at:

·      Durmus, D. (2022). Correlated color temperature: Use and limitations. Lighting Research & Technology, 54(4), 363-375.

·      Esposito, T., & Houser, K. (2022). Correlated color temperature is not a suitable proxy for the biological potency of light. Scientific Reports, 12(1), 20223.

I personally do not think the use of first-person pronouns is appropriate for academic writing.

Figure 2: what do the errors bars represent? St.dev, error of the mean?

Figure 3 provides good insight. I’m sure other readers will appreciate these types of data display.

I must have missed, but I didn’t see any effect sizes reported.

The limitations section in the paper is commendable as it highlights potential constraints and boundaries of the study's findings. By acknowledging these limitations, the authors demonstrate intellectual honesty and provide valuable context for interpreting the results.

The language and style are generally good, but a final review by a native English speaker or proof-reader would improve language quality.
